# Intelligent Packaging Systems with Anthocyanin: Influence of Different Polymers and Storage Conditions

**DOI:** 10.3390/polym16202886

**Published:** 2024-10-14

**Authors:** Leandro Neodini Remedio, Carolina Parada Quinayá

**Affiliations:** 1Faculty of Animal Science and Food Engineering, University of São Paulo USP, Av. Duque de Caxias Norte 225, Pirassununga 13635-900, SP, Brazil; 2Bioengineering and Chemical Engineering Department, Universidad de Ingenieria y Tecnologia UTEC, Jr. Medrano Silva 165, Lima 15063, Peru; dparada@utec.edu.pe

**Keywords:** anthocyanins, indicator films, stability, intelligent packaging

## Abstract

With the aim of meeting the growing demand for safe food, intelligent packaging has emerged, which monitors the conditions of the food and informs the consumer about its quality directly at the time of purchase. Among intelligent packaging options, colorimetric indicator films, which change color in response to changes in the food, such as the release of volatile compounds, have been widely studied. Among them, pH indicator films composed of dyes sensitive to small variations in the pH value of the food surface have received greater attention in recent years. Anthocyanins, which are natural pigments, have stood out as one of the most commonly used sources of dyes in the production of these indicator films. In this context, the present review aims to present an updated overview of research employing anthocyanins in indicator films, including their stability under different storage conditions, the influence of different polymers used in their production, and alternative techniques for maintaining stability.

## 1. Introduction

The main purpose of food packaging is to preserve food quality and safety by maintaining its nutritional values and freshness, while also ensuring safe storage during distribution. In recent years, there has been a significant increase in research related to the development of new packaging technologies capable of detecting small variations during product storage or transport. This need is also evident in consumers, who are increasingly demanding products with superior qualities and tests with greater sensitivity to indicate the beginning of a potential product deterioration process [1].

According to Oliveira Filho et al. [2], packaging plays several crucial roles in food products, such as safeguarding their contents from contamination and deterioration, facilitating their transportation and storage, and providing standardized measures of their contents. Its main functions are defined as containment, protection, convenience, and communication.

In response to the increasing demand for safe food products, smart packaging has emerged, aiming to monitor food conditions and promptly inform consumers about the quality of packaged foods at the time of purchase [3].

Interest in the development of these smart packaging solutions has grown in recent decades to track food quality, with colorimetric indicators receiving particular attention. These indicators provide direct and visual information to consumers about changes in food quality [4]. Unlike traditional packaging, smart films containing colorimetric indicators aim to reflect the freshness of food by detecting surface alterations, causing them to change color and instantly signaling to the consumer that the product is not suitable for consumption. Colorimetric indicator films that change color in response to pH variations have been extensively studied in recent years [5,6,7,8,9].

These new smart pH indicator films have garnered attention due to their properties such as providing a quick response, preserving food integrity, convenience, and the ease of freshness detection with the naked eye. They are generally composed of dyes which are sensitive to small variations in the pH value of the food surface [1].

Natural pigments have been the focus of recent research due to being considered environmentally friendly and safe for consumption compared to synthetic dyes. Among these natural pigments, anthocyanins represent an important class because they are water-soluble, exhibit a wide range of responses to different pH values and variations, and can be obtained from various sources [4].

Among naturally occurring pigments, anthocyanins have stood out as being readily available in nature. Numerous studies have utilized anthocyanins to produce different intelligent packaging films or smart labels to inform consumers about the quality and freshness of food. Thus, this review aims to provide an updated overview of current studies employing anthocyanins from different sources in indicator films with real applications in food products. Additionally, it will explore their stability under different storage conditions, long-term effects, the influence of different polymers used in their production, and alternative techniques for maintaining stability.

## 2. Utilization of Anthocyanins as Intelligent Indicator Films

When it comes to intelligent indicator packaging, there is a strong preference for natural compounds. However, these packaging materials must have the ability to indicate to consumers that there has been a change in the product which can then lead to a real judgment of whether the product has indeed deteriorated or is unfit for consumption. The main feature of intelligent packaging lies in its ability to communicate the different characteristics of products to consumers, regardless of whether the consumer has interacted with them or not, with freshness being one of the most important and concerning attributes for consumers and retailers as it can indicate and ensure the quality and safety of a food product [5].

Intelligent indicator packaging relies on qualitative and quantitative changes in one or more factors, leading to visual responses that directly indicate variations in compounds related to food deterioration. Examples of such packaging include pH variation indicators [6,7,8,9,10], variations in or the production of volatile compounds [8,9,11,12,13], and indicators of gas presence [14,15,16,17,18].

The most commonly found application involves incorporating films with colorimetric properties into the packaging. However, their contact with the food varies depending on the function performed by the packaging and whether it involves direct contact with the food or not (Figure 1).

### 2.1. pH Indicators

Many studies have demonstrated the addition of anthocyanins to pH indicator packaging which are commonly used to monitor the quality and freshness of foods, as the deterioration of many products correlates with changes in pH values [19]. Several recent studies have shown the possibility of using intelligent packaging as an indicator of pH variations in foods (Table 1).

Recently, many studies have demonstrated the effectiveness of using anthocyanins in pH indicator packaging to monitor food quality and freshness, with the deterioration of many foods being correlated with pH variations, as can be seen in Table 1.

Qin et al. [29], studying the development of chitosan-based films combined with anthocyanins obtained from purple corn extract, observed notable changes when the films applied to different buffer solutions, varying from pH 3 to 10. These changes were attributed to the structural transformation of anthocyanins. Purple colorations were observed at a pH ranging from 3 to 6, while a blue coloration was observed when the pH was between 7 and 10, corresponding to carbinol pseudobases and the quinoid forms of anthocyanins. According to the authors, the developed films can be easily used to monitor the freshness of packaged foods. Similarly, Yong et al. [6] found color variations in their chitosan-based films combined with purple sweet potato extract rich in anthocyanins when they were immersed in buffer solutions ranging from pH 3 to 10. The observed colors of the films varied from reddish pink at a pH between 3 and 6, to brown-purple at a pH between 7 and 8, and green at pH 9 and 10. Furthermore, there was a difference in the degree of color change in the films depending on the concentration of extract added, with higher concentrations showing greater visual differences.

pH indicators have been applied to assess the quality of various foods, with studies primarily focusing on pork, poultry, fish, shrimp, and milk. For pork meat, its surface pH value is a crucial factor and can easily indicate their freshness status. Values ranging from 5.18 to 6.12 suggest that the meat is still fresh and suitable for consumption. pH values between 6.13 and 6.16 indicate a slight decrease in freshness, and its consumption is at the acceptable limit, and pH values of 6.18 and above already demonstrate the onset of product deterioration [35].

Choi et al. [7] applied pH indicator films, produced based on agar and potato starch with purple sweet potato extract rich in anthocyanins, to pork samples under accelerated storage conditions (at room temperature −25 °C). It was observed that the pH values ranged from 5.82 at the beginning of the analyses, increasing to 6.28 after 20 h of storage. After 48 h, the samples reached their maximum point of deterioration, reaching pH 7.42. According to the authors, this increase is due to the formation of deteriorating compounds such as ammonia and amino acid complexes, resulting from the decomposition of substances such as lipids and proteins. Correlating these results with the observed color values of the films (initially red when pork sample was fresh), the authors observed significant variations in the L*, a*, and b* parameters, ranging from 64.25, 12.23, and 6.72 (L*, a*, and b*, respectively) for fresh meat (pH 5.78) to 73.70, 3.53, and 12.39 (L*, a*, and b*) after 20 h of storage (pH 6.28). Finally, after 48 h of storage, the samples had a pH of 7.46, and the films turned completely green, presenting L*, a*, and b* values of 81.55, 1.50, and 8.29, respectively.

Fresh cow’s milk, which is an amphoteric solution due to the presence of citric acid, phosphates, and casein, has a pH value between 6.6 and 6.8 and pH is one of the most commonly used parameters for analyzing milk quality [26]. Using cellulose and chitosan-based pH indicator packages with carrot anthocyanins, Tirtashi et al. [26] observed that raw milk samples had a pH value of around 6.6. After 48 h of storage at 20 °C, the pH values dropped to 5.7. Accompanied by physicochemical and microbiological variations, the authors observed changes in the color of the indicator films, changing from blue to pink-violet after 48 h of storage, clearly indicating milk deterioration, and with a color difference value (ΔE) of 29.28 after 48 h. Generally, the ΔE value is applied to analyze the colorimetric parameters of the indicator film, with color changes being visually perceptible when the ΔE value is greater than 5.

pH indicator packaging can also be applied to dairy products to determine proper pasteurization and sterilization processes. Applying chitosan-based films, PVA, and anthocyanins from red cabbage directly onto pasteurized milk samples, stored at 20 °C for a period of 4 days, Pereira Jr. et al. [20] detected a visible change in their coloration. According to the author, milk begins to decompose when not stored under ideal refrigeration conditions (7 °C), causing variations in pH and reducing the milk’s stability. In the first day, with a pH value of 6.7, the films exhibited a dark gray coloration. As the pH decreased to 5.0, the indicator films became lighter. When the milk reached a pH of 4.6, already considered a deteriorated sample, the films showed a light pink coloration. The observed ΔE values ranged from 21.07 to 36.85, which could be observed with the naked eye.

pH changes can also be utilized to demonstrate the degradation of fish and other types of seafood. During protein degradation, the formation of volatile alkaline nitrogen increases the pH of foods. When combined with smart pH indicator packaging, this factor enables the determination of the precise moment when seafood is no longer fit for consumption [8]. For such food items, the pH values gradually increase with extended storage time, resulting in significant variations in color in anthocyanin-rich films due to their sensitivity to pH [32]. In a sealed package, the pH of the environment increases as volatile nitrogenous compounds are retained [19].

Wu and Li [8] observed color variations in pH indicator films made from CMC and PVA with anthocyanins from carrot which were applied to carp filets and stored at 4 °C for 10 days. Due to the low storage temperature, the films exhibited color variations after a longer period, with a pink hue being evident until day 6 of storage, followed by a transition to brick red after 8 days, and finally a reddish-brown coloration after 10 days of storage. These color changes were positively correlated with the fish freshness index results. According to the authors, these colorations were associated with fresh meat, semi-fresh meat, and spoiled meat, respectively, indicating their significance in evaluating storage conditions and fish preservation. Similar results were also reported by Amaregouda et al. [32], Ma et al. [19], and Silva-Pereira et al. [21], who evaluated the use of pH indicator films supplemented with anthocyanins for samples of different fish species.

Similarly to fish, the freshness of shrimp can also be determined using pH indicator films. When stored at room temperature (25 °C), shrimp begin to change their coloration from gray to red after 36 h, exhibiting a more intense red coloration with increased storage time, indicating their deterioration. However, for samples stored under refrigeration (4 °C), these variations do not appear until 72 h of storage, making it difficult to detect potential changes in product freshness [23].

Bao et al. [27] produced pH indicator films based on potato starch with anthocyanins from blueberry extract and applied them to shrimp samples to monitor their freshness when stored in a refrigerator at 4 °C for 36 h. The pH values observed during the storage period increased from 6.91 to 7.41, significantly altering the freshness of the samples. The color difference value of the films (ΔE) was 10.12 between the initial day and the critical deterioration point of the shrimp samples. According to the authors, the color change can be intuitively perceived by the naked eye, and the freshness monitoring test indicated that the shrimp had deteriorated completely after 36 h of storage.

### 2.2. Volatile Compound Indicators

As with pH variation indicators, anthocyanins can also be employed in smart packaging or indicator films to show the release of volatile compound, with the main difference being the absence of direct contact between the film and the food. The bacteria that are typically responsible for food deterioration produce metabolites such as volatile compounds, which accumulate inside the packaging and consequently induce a color change in the indicators, informing the consumer about the food’s condition [36]. Additionally, as the freshness of protein-rich foods diminishes, the release of volatile basic nitrogen can generate OH– groups under certain humidity conditions, leading to an increase in the environment’s pH [37].

Recently, many studies have been conducted on indicator packaging in which anthocyanins are applied to demonstrate the release of Total Volatile Basic Nitrogen (TVB-N) derived from food decomposition. TVB-Ns, basic gasses formed according to biochemical and microbial activity, such as trimethylamine, are associated with the unpleasant taste of spoiled fish, leading to sensory rejection [36].

Foods used to test volatile compound indicator films include fish, pork, chicken, and shrimp, among others. Due to the high solubility of anthocyanins, indicator films are typically added to the top of food packaging with a high water content, without direct contact with the food. This same process is used for gas indicator packaging, and several recent studies on this can be found in the literature (Table 2).

Kim et al. [38] applied gelatin and agar-based films combined with anthocyanins from flower petals to determine the freshness of shrimp stored at 4 °C over 9 days. After 6 days, the initial TVB-N value in the shrimp samples increased from 7.7 to 20.7 and reached 34.5 mg/100 g on day 9 of storage, while the pH increased from 6.8 initially to 7.0 and 7.6 during the same period. The films initially exhibited a pink coloration, changing to green on the last day of the trial. According to the author, shrimp is considered fresh when the TVB-N value is below 20 mg/100 g, while values exceeding 30 mg/100 g are considered spoiled. Similarly, Dong et al. [39] also assessed the freshness of shrimp samples when packaged with agar, chitosan, and anthocyanin-based films derived from sweet potatoes, refrigerated at 4 °C over a 5-day period. On the fourth day of analysis, the shrimp samples exhibited a TVB-N value of 19.88 mg/100 g, approaching the consumable limit (20 mg/100 g) according to food standards, indicating that from this point onward, the shrimp would no longer be suitable for consumption. In comparison to the pH results of the samples, the authors noted a decline in the pH value on the first day. This phenomenon occurs due to the shrimp’s glycogen undergoing glycolytic processes following oxygen deprivation after death, resulting in the production of pyruvate and a subsequent pH reduction. However, shortly afterward, the shrimp proteins decomposed to produce an alkaline substance, resulting in an increase in the pH. This rise in pH caused a color variation, shifting from reddish pink to gray tones after 5 days, clearly indicating to consumers the deteriorated state of the product, rendering it unfit for consumption.

Other foods, such as pork, have also drawn attention, as their deterioration is directly related to microbial reproduction. The microbiological degradation of proteins and fat in pork produces a large number of volatile compounds, which lead to food spoilage and a consequent variation in its pH value [22].

Corn starch-based films and purple corn powder rich in anthocyanins showed a change in their coloration from pink to greenish gray as the TVB-N values of pork samples increased from 2.70 mg/100 g to 14.03 mg/100 g after 23 h at 25 °C [37]. This visible change in the color of the film to the human eye can signal that the pork is about to spoil, aiding in food safety enhancement.

With the growth of microorganisms and biochemical reactions, the freshness of pork decreases, and proteins, fats, and carbohydrates are decomposed to produce volatile nitrogen. Variations in TVB-N values were reported in another study monitoring the freshness of pork, which increased from 3.52 to 48.73 mg/100 g after 60 h of storage. These variations resulted in changes in the colorations of films based on cassava starch, PVA, and grape skin, shifting from pink to yellow, indicating that the meat was completely deteriorated and unfit for consumption. Additionally, an increase in the color difference (ΔE) was observed, rising from 15.36 to 45.72, representing a change noticeable to the naked eye [40].

Like pork and shrimp, fish is a typical perishable material. The deterioration of most fresh fish and fish-based products is primarily due to the action of microorganisms and biochemical reactions, and the resulting increase in TVB-N levels is related to the degree of protein degradation caused by the growth of deteriorating bacteria and the action of endogenous enzymes [41].

**Table 2 polymers-16-02886-t002:** Smart films indicators of volatile compounds and gasses based on different sources of anthocyanins and polymers for food application.

Source of Anthocyanin	Anthocyanin Concentration	Film-Forming Polymer	Type of Indicator	Food Application	References
Red cabbage	0, 20, 40, and 60 mg/100 g solution	PVA and CMC	pH variation and Total Volatile Basic Nitrogen (TVBN) release indicator	Pork meat	[22]
	0.7 g/100 mL solution	Agar and methylcellulose	Total Volatile Basic Nitrogen (TVBN) release indicator	Chicken breast	[11]
	2.5 g/100 g polymer	Gelatin	Total Volatile Basic Nitrogen (TVBN) release indicator	Fish	[13]
Purple sweet potato	0.2, 0.4, 0.6, 0.8, and 1.0 g/100 g polymer	Sodium alginate	NH_3_ presence indicator	Chicken breast	[16]
	1.5 g/100 g polymer	Agar, chitosan and sodium alginate	Total Volatile Basic Nitrogen (TVBN) release indicator	Fresh shrimp	[39]
	-	Sweet potato powder and sodium alginate	Total Volatile Basic Nitrogen (TVBN) release indicator	Fresh shrimp	[12]
Black carrot	6 mg/mL solution	Bacterial cellulose nanofibers	pH variation and Total Volatile Basic Nitrogen (TVBN) release indicator	Fish filet (rainbow trout and common carp)	[36]
	45 mg/100 mL solution	PVA and CMC	pH variation and Total Volatile Basic Nitrogen (TVBN) release indicator	Fish (carp)	[8]
Blueberry	0.1 g/100 g polymer	Potato starch	pH variation and Total Volatile Basic Nitrogen (TVBN) release indicator	Fresh shrimp	[27]
	0.15 mg/mL solution	PVA and corn starch	CO_2_ presence indicator	Mushrooms	[14]
	0.3 g/100 mL solution	PVA	NH_3_ presence indicator	Pork meat	[15]
Black corn	21 mg/100 g black corn powder	Black corn starch and κ-carrageenan	Total Volatile Basic Nitrogen (TVBN) release indicator	Pork meat	[37]
Mulberry	15, 30, and 45 mg/100 g solution	Gelatin and PVA	Total Volatile Basic Nitrogen (TVBN) release indicator	Fish (carp)	[41]
	4, 8, 12, and 16 g/100 g polymer	Chitosan, PVA, sodium alginate and pullulan	Total Volatile Basic Nitrogen (TVBN) release indicator	Chinese mitten crab	[42]
Black soybean	0.02–0.30 g/100 g polymer	Sodium alginate	Total Volatile Basic Nitrogen (TVBN) release and Hydrogen sulfide indicator (H_2_ S)	Pork meat	[18]
Grape skin	10 g/100 g cassava starch	cassava starch and PVA	Total Volatile Basic Nitrogen (TVBN) release indicator	Pork meat	[40]
Roselle	30, 60 and 120 mg/100 g starch	Starch and PVA	Total Volatile Basic Nitrogen (TVBN) release indicator	Fish filet	[4]
	2.5/100 g polymer	PVA, chitosan and starch	Total Volatile Basic Nitrogen (TVBN) release indicator	Pork meat	[43]
	10 g/100 g polymer	Gelatin and agar	Total Volatile Basic Nitrogen (TVBN) release indicator	Fresh shrimp	[38]
	10 mL/100 g polymer	Agar and PVA	NH_3_ presence, and release of trimethylamine (TMA) and dimethylamine (DMA) indicator	Salmon	[17]
Eggplant	2 and 4 g/100 g polymer	Chitosan	pH variation and Total Volatile Basic Nitrogen (TVBN) release indicator	Pork meat	[44]
Plum peel	5, 10 and 15 g/100 g polymer	Sodium alginate and gelatin	pH variation and Total Volatile Basic Nitrogen (TVBN) release indicator	Chicken breast	[9]

Indicator films based on gelatin containing 0.28% anthocyanins from purple cabbage extract have been applied in the preservation of fresh fish samples stored at 4 °C. Initially, the TVB-N values were 13.3 mg/100 g, with the films exhibiting a pink coloration. After 4 days of storage, these values increased to 23.53 mg/100 g, suggesting that the fish samples were beginning to deteriorate. By day 6, the TVB-N values reached nearly 40 mg/100 g, and the color turned blue, indicating a visible color difference in the films and confirming their function in monitoring the quality of fresh fish [13].

When indicator films are applied to chicken samples, studies indicate that, when the TVB-N values are below 15 mg/100 g, the samples are considered fresh, becoming less fresh when these values range between 15 and 30 mg/100 g, and deteriorated for values above 30 mg/100 g [33]. For chicken samples monitored with agar, methylcellulose, and red cabbage extract films, Hashim et al. [11] observed that the initial level of freshness, determined by the concentration of TVB-N, was 8.8 mg/100 g, with the indicator films exhibiting a pink color. The TVB-N value increased to 20.6 mg/100 g at 18 h (dark pink) and 26.3 mg/100 g at 27 h (green color) as the food deteriorated. These results were within the specified limit (25–28 mg/100 g) for the initial deterioration point of chicken meat.

Chen et al. [9], monitoring the freshness of chicken using indicator films based on sodium alginate, gelatin, and anthocyanins present in plum skin extract, found chicken TVB-N values ranging from 7.73 to 34.58 mg/100 g during a 10-day storage period, and pH ranging from 5.82 to 6.84 during the same period. The color of the indicator film changed from red and pink-orange to green. As the TVB-N value and pH increased, the color of the composite film gradually darkened, being affected by volatile ammonia created by the breakdown of fat, protein, and carbohydrates, suggesting that the studied film and extract can be used to monitor the freshness of chicken breasts.

### 2.3. Gas Indicators

Some smart packaging can indicate the release of specific gasses. Anthocyanins, when in contact with gasses released due to food decomposition, such as ammonia (NH_3_), change their coloration due to the creation of an alkaline environment caused by this gas [15]. Essentially, ammonia first diffuses into the indicator films, and then hydrolyzes to produce hydroxyl ions, which create an alkaline environment. As a result, the structure of the anthocyanins is converted to chalcone, causing the indicator films to change color [43].

CO_2_ is a gas that is constantly released during the degradation of mushrooms [14]. This occurs because, during the respiration process of mushrooms, the concentration of CO_2_ increases while that of O_2_ decreases, leading to a decrease in the pH within a package, which can also be related to the deterioration of the sensory texture of the mushrooms [45].

When evaluating the freshness of mushrooms stored for 6 days at 20 °C in packaging containing smart films based on PVA and corn starch, supplemented with anthocyanins from blueberry extract, Liu et al. [14] observed changes in film color perceptible to the naked eye. With increased storage time, the color varied from purple on days 0 and 1 to reddish-brown on days 2 and 3, and pink after day 4. The ΔE value during this period was 7.45 at the end of storage compared to the first day, which was primarily due to the CO_2_ released by the mushrooms, altering the structure of the anthocyanins.

Hydrogen sulfide (H_2_S) is another type of gas produced during the degradation of meat products, generated through the breakdown of proteins and sulfur-containing amino acids by microbial enzymes. In this process, the surface of the films absorbs water molecules from the environment, and subsequently, H_2_S reacts with these molecules, producing hydrosulfuric acid, along with compounds such as methionine, cysteine, and others. This reaction alters the pH of the system, leading to changes in its color [18,46].

Shi et al. [18], while monitoring the degradation of pork over 5 days at 25 °C, observed an increase in the concentration of H_2_S in the system from 11.24 mg/100 g at the beginning of the analysis to 25.22 mg/100 g after 2 days, reaching values close to 50 mg/100 g after 5 days of storage, indicating complete deterioration. The coloration of the films based on sodium alginate added with anthocyanins from black soybeans ranged from light brown to a yellowish brown hue, being visibly perceptible to the naked eye according to the authors.

When PVA-based indicator films added with blueberry extract were exposed to NH_3_, Zhang et al. [15] observed significant color variations. After just a 12 min exposure period, the films exhibited color changes ranging from purplish gray to blue and then to bluish green. The color parameter that underwent the most significant alteration was the a* parameter, which decreased from 24.92 to 5.26, and the color difference (ΔE) ranged from approximately 19 to values above 30, indicating a visually significant alteration that can be observed by a potential consumer.

Smart packaging indicators based on cellulose nanofiber combined with red cabbage extract, exposed to NH_3_ for 30 min, showed significant variations in the color parameters L*, a*, and b*. Before exposure to NH_3_ gas, the samples had a dark brown color, shifting to more yellowish hues after exposure. The NH_3_ present in the environment interacts with the H_2_O in the films, generating NH_3_:H_2_0, which hydrolyzes to release NH4^+^ and OH^−^, causing an increase in pH and consequently a change in the color of the films [24].

Xiaowei et al. [17] analyzed the colorimetric response of anthocyanins from nine different raw materials when exposed to fresh salmon samples over a 12-day period. According to the authors, sensory changes in the salmon samples are difficult to discern with human senses, yet are easily detected by anthocyanins, providing reliable recognition. Increased exposure time to NH_3_ led to heightened color intensity of the anthocyanin samples, confirming the feasibility of using anthocyanins as gas-sensitive materials to monitor the salmon decomposition process, with roselle extract exhibiting the most perceptible color variation compared to other raw materials. When incorporated into agar and PVA-based films and analyzed in salmon samples, the anthocyanins exhibited color variations (ΔE) ranging from 5.94 in the early days to 10.87 after 10 days of storage, transitioning their color from red to green. According to the authors, an increase in the ΔE value above 10 indicates a perceptible variation visible to the naked eye as it enters a new color space, suggesting that their indicator films can effectively predict the chemical quality of salmon samples.

## 3. Stability of Anthocyanins in Smart Indicator Films

The stability of anthocyanins’ color is an important parameter for evaluating the performance of smart indicator packaging color in relation to food freshness, mainly due to the low stability of anthocyanins [11]. Their stability is directly related to their chemical structures, where acylated anthocyanins, due to their intramolecular copigmentation effect, are more stable than non-acylated ones [12,47]. The thermal behavior of anthocyanins may differ in more complex food matrices, for example, in foods enriched or fortified with anthocyanins, due to interactions with some nutrients such as proteins and polysaccharides that can stabilize these pigments [48].

Anthocyanins undergo structural transformations, and consequently drastic changes in color, with changes in pH. At pH 3 or lower, the color of anthocyanins ranges from orange to bluish red, depending on their chemical structure, and is predominantly due to flavilium cations [47]. Due to their transformations at different pH values, their application in food systems has been more focused on acidic foods to ensure the predominance of these cations [47]. However, a point that contributes to the greater color stability of pH variations is the ability of acylated anthocyanins, especially di- or poly-acylated ones, to resist hydration, exhibiting coloration in weakly acidic, neutral, and slightly alkaline solutions [47].

In some cases, the indicative packaging fails to exhibit the necessary performance to track the stage of deterioration due to its color variation, which may be related to two points: the characteristics of the anthocyanins present in the raw material used, influencing the type of indicator, and the way this indicator is used [26].

To monitor rapidly deteriorating foods, studying the color stability of indicator films is crucial. Several studies in the literature have reported that anthocyanins are easily affected by factors such as temperature, lighting, and pH.

### 3.1. Temperature

The stability of anthocyanins can be easily altered by temperature, as the chalcone structure present in anthocyanins is formed when the ambient temperature approaches value close to 60 °C, resulting in a decrease in pigments with prolonged heating at a constant temperature [49]. Recently, several studies on the stability of anthocyanins in smart films have been published in the literature. An example of a technique used in stability studies is the use of temperature variations, where samples are subjected to storage at different temperature conditions for long periods, and the color variation (ΔE) is evaluated over time. Hashim et al. [11] assessed the color stability of their films based on agar, methylcellulose, and cabbage extract at 4 °C and 25 °C with 50% relative humidity for a period of 16 days. During the first 10 days, the films stored at 4 °C had only slight variations in ΔE, rapidly increasing to 4.7 after 16 days, compared to the films stored at 25 °C, which had a ΔE of 3.7 in the first 10 days and 5.3 at the end of the 16-day storage period. According to the authors, the porosity of the films may have allowed for increases in oxidation, moisture, and temperature, resulting in a reduced color.

Similarly, Huang et al. [42] investigated the color stability of films based on gellan gum combined with anthocyanin-rich blackberry extract over a period of 14 days when subjected to 4 °C, 25 °C, and 37 °C with a constant relative humidity of 75%. Overall, it was observed that the stability of the films was better when stored at 4 °C, as higher temperatures accelerated the degradation of anthocyanins, causing the diphenyl benzopyran cations to react towards the pseudobase and chalcone, resulting in the fading of the anthocyanins. According to the authors, the main reason for the good stability of the films is that the polysaccharide used is a good oxygen barrier material, which can effectively separate the oxygen degradation effect on the internal layer of anthocyanin, thereby greatly improving the film’s stability, making it more suitable for application as a pH indicator in smart packaging. The observed ΔE values over the 14-day period ranged from 0 to 0.5 for films subjected to 4 °C and from 0 to 4 when subjected to 25 °C, and when stored at 37 °C, the observed ΔE values ranged from 0 to 15, showing the highest variation observed among the samples.

Before they can be used in the industry as pH-sensitive smart packaging, it is necessary to assess the ability of these packages to maintain their colors until the end of their shelf life, thus providing reliable visual feedback to consumers. Wu and Li [8] evaluated the stability of films at 4 °C and 25 °C over 20 days. The ΔE values stayed below 5, but increased above 5 after 16 days at 25 °C.

For films based on purple sweet potato starch and sodium alginate, combined with anthocyanins from the same raw material used in the production of the films, Li et al. [12], analyzing the color stability in films stored at 4 °C and 25 °C for 30 days, did not observe significant changes in the ΔE values of their samples. The authors observed only minor alterations, such as an ΔE of less than 2 for the samples subjected to 4 °C and less than 3 when subjected to 25 °C, demonstrating good stability under different storage conditions.

Bao et al. [27] reported a better color stability at 4 °C than at 25 °C, with ΔE values of 3.36 and 4.04, respectively, after 14 days. As reported by other authors, these values are below what can be observed by the human eye (ΔE > 5), making them excellent materials for use as pH indicators for smart packaging. Their good stability occurred because higher concentrations of chondroitin sulfate cause greater hydrogen bond interactions, making the copigmentation effect between anthocyanins and chondroitin sulfate more efficient in maintaining the color of the films over time.

### 3.2. Light

Light can be an important factor in the stability of anthocyanins for use in pH indicator smart packaging. Therefore, some recent studies have focused on determining the stability of films when subjected to the same temperature conditions but varying the presence and absence of light during storage.

Bao et al. [27] evaluated color stability under light and dark conditions. Films stored in the dark exhibited a high stability with ΔE values below 3.16 after 14 days, while those exposed to light showed increased ΔE values of around 4.29, demonstrating that light affects anthocyanin stability. Although the results are close and within a range that is perceptible to the human eye, it can be demonstrated that the presence of light affects the stability of anthocyanins.

Similarly, Wu and Li [8] tested their films under light exposure. The presence of light caused a slight influence on the color parameters of the films. Over 20 days, the films exhibited a ΔE of around 2 when stored without light and values above 3 when exposed to light. For films with polyphenols, the ΔE values were around 6 in the dark and over 7 with light, showing that light exposure affects long-term color stability.

Although some authors have been able to observe small but significant differences in the color variation values (ΔE) when subjecting films added with anthocyanins to light presence and absence, few studies are found containing real applications in foods, making it a promising area of study.

## 4. Production and Characterization of Indicator Films with Anthocyanins

Smart freshness indicators for food comprise a polymeric base and a pH-sensitive dye, and various polymers, dyes, and preparation methods have already been evaluated. The ability of an indicator film to differentiate food deterioration stages is affected by the compatibility of the polymeric base, the type of dye, and the preparation method [3].

The casting method is the most common method for the production of films in the literature, with tape casting being a variation of this method. Tape casting has the ability to produce films with more rigorous thickness controls, ensuring a greater homogeneity in the films produced [50].

In the casting technique, initially, a filmogenic solution is formed containing the desired polymer, to which an additive containing a specific property is added, for example, an extract from a type of fruit with high concentrations of anthocyanins. After homogenization, the final solution formed is poured into a Petri dish, and the mass added must be controlled to ensure a consistent film thickness. Finally, the films are dried in an oven with controlled temperature and time parameters. For the tape casting technique, the process is practically the same, except that instead of pouring the solution into Petri dishes, it is applied onto a spreader (tape caster), where the thickness can be controlled to achieve a consistent thickness. Subsequently, the solution undergoes drying to initiate its characterization (Figure 2).

According to Ockun et al. [51], the quality of the films depends on the viscosity of the polymeric matrix, which plays an important role when using the casting or tape-casting method. In the casting technique, solutions with high viscosities are not commonly used, resulting in thinner and more flexible films. In contrast, for the tape casting technique, more viscous solutions are preferred because, upon deposition, the thickness remains unchanged, ensuring better control.

Preparing smart films based on chitosan, methylcellulose, and anthocyanins for monitoring the freshness of fish filet samples using the casting technique, Gasti et al. [52] created films with a good formation and a controlled thickness of 27 μm. According to the authors, film thickness is a crucial parameter for evaluating characteristics such as mechanical properties and light barrier properties.

Forghani et al. [53], produced colorimetric indicator films based on PVA and κ-carrageenan combined with anthocyanins from *Centaurea arvensis* using the casting technique instead of electrospinning to monitor food freshness using pH variation, and showed that the mechanical and water barrier properties of the films were influenced by the preparation method. According to the authors, the differences may be related to the faster production of fibers using the electrospinning method compared to the casting method, which leads to more limited interactions between κ-carrageenan and PVA.

The physical appearance of films is an important parameter that determines their effectiveness in becoming a suitable candidate for packaging applications. Preparing pH-sensitive films based on starch and carbon combined with anthocyanins to monitor pork degradation using the casting technique in their production, Koshy et al. [54] observed the formation of homogeneous films without visible cracks or bubbles. These films could be easily removed from their formation plate and exhibited a reddish purple color.

In order to obtain a final product with good characteristics and provide a clear and objective response to the consumer, indicating whether the product is suitable for consumption, it is necessary to evaluate which polymer should be used in its production, what characteristics are desired, and how it responds in real application conditions.

### 4.1. Types of Polymers Used in pH Indicator Films

Several polymers can be used in the production of pH indicator packaging, such as polyvinyl alcohol (PVA) [8,15,17,19,22,41], potato starch [7,27], corn starch [14,21,37], cassava starch [40], chitosan [6,21,25,26,31,32,34], agar [7,11,17,38,39], carboxymethyl cellulose (CMC) [8,22,30,33], and gelatin [9,13,38,41], among others.

Water-based polymers have been the most commonly used, primarily because the hydro-solubility of anthocyanins facilitates their incorporation into these polymers. To ensure the stability of anthocyanins, polymers must be stable at low pH, colorless to avoid interactions with anthocyanins, sensitive to moisture to allow correct film operation and indication, and possess suitable mechanical properties. Typically, multiple polymers are used in combination to ensure better properties [5]. Figure 3 shows some of the polymer combinations found in the literature, with the most commonly used being PVA, starch, and chitosan.

Liu et al. [14] produced pH indicator films combined with anthocyanins from blueberry, based on PVA and starch, and observed appropriate mechanical properties, moisture resistance, light barrier capacity, and thermal stability. These results were attributed to the strong interaction between the film’s components, mainly related to the hydrogen bonds formed.

Evaluating films based on chitosan and starch, supplemented with cabbage extract rich in anthocyanins, Silva-Pereira et al. [21] found that besides the components used in the film production not reacting with each other, the blend exhibited good thermal stability and a uniform morphology, with a continuous polymeric matrix and no irregularities. The films showed potential for use as indicators of food deterioration as they were able to detect pH variations.

Other types of blends can be used to improve the final characteristics of films, such as PVA and chitosan-based films with blackberry extract as a source of anthocyanins for the production of pH indicator films [19]. X-ray diffraction and FT-IR results for the produced films revealed hydrogen interactions between the components, and scanning electron microscopy analysis showed compact structures formed by the blend. Furthermore, the addition of 6% chitosan to the films was sufficient to maximize their mechanical properties, making them suitable for use as smart packaging in the food industry.

### 4.2. Physical–Chemical Characteristics

Some physical–chemical analyses are essential when producing films with potential applications in the food industry, especially if they are supplemented with anthocyanins, as their structure can directly affect factors such as moisture content, water solubility, mechanical properties, and water barrier capacity, among others.

Liu et al. [22] analyzed the effect of adding cabbage anthocyanins to PVA- and CMC-based pH indicator films and observed that the addition of anthocyanins caused a reduction in water solubility values from 55.8% to around 40.0%, possibly due to an interaction between the hydroxyl groups of anthocyanins and the PVA matrix, forming hydrogen bonds. However, increasing the concentration of anthocyanins added caused an increase in the solubility of the films, possibly due to the formation of a three-dimensional network between the anthocyanin molecules and the matrix.

Despite not affecting the thickness of a film, the addition of anthocyanins can significantly alter its mechanical properties, reflecting the formation of molecular crystals and the density of film molecules. Studying the incorporation of different concentrations of cabbage powder containing anthocyanins into chitosan-based films, Chen et al. [55] observed that the addition of anthocyanins caused a significant increase in its mechanical strength and flexibility. When 1.2% of cabbage powder was added, there was a 13.3% increase in the tensile strength of the films, possibly due to the formation of hydrogen bonds between the hydroxyl groups and the amino groups present in chitosan and the polyphenols of anthocyanin, and a 23.33% increase in their elongation, due to the plasticizing effect of the anthocyanins source used.

In the development of cellulose nanofiber-based films with anthocyanins from purple sweet potato and oregano essential oil as a pH indicator and antimicrobial, Chen et al. [56] observed that the addition of anthocyanin and oregano essential oil improved the light barrier properties of the films against ultraviolet light while exhibiting antimicrobial activity against *E. coli* and *Listeria monocytogenes*. Moreover, the films responded sensitively to pH changes, showing a noticeable color variation from red to yellow as the pH ranged from 2 to 12, thereby enabling their use for monitoring changes in food quality during storage, acting as intelligent pH indicators.

In colorimetric indicator films based on PVA and gelatin incorporated with blackberry extract, Zeng et al. [41] observed notable effects. They found that the addition of this extract at a concentration of 45 mg/100 mL led to a decrease in tensile strength values from 30.8 to 21.0 MPa, and an increase in elongation values from 589.22% to 905.86%, with no significant differences observed in film thickness values. Apparently, the blackberry extract contains numerous hydroxyl groups that can act as plasticizers, enhancing the macromolecular mobility of the PVA/gelatin blend, resulting in a less dense polymeric network due to the reduction in intermolecular forces.

### 4.3. Responses under Real Storage Conditions

Due to microbial growth or chemical changes in the food, resulting in the release of gasses, pH indicator films aim to provide instant details to consumers regarding the product’s quality, informing them of whether its consumption is safe [57].

During the deterioration of certain products such as pork, shrimp, among others, proteins are decomposed into peptides and amino acids, which convert into gasses such as CO_2_, ammonia, dimethylamine, and trimethylamine, known as TVB-N, through biochemical and microbiological reactions, resulting in a pH variation that can be easily detected by indicator packaging [58].

In Table 3, the response obtained from pH indicator films based on different sources of anthocyanins when applied to foods can be observed.

Dong et al. [39], producing multifunctional smart films based on agar and sodium alginate supplemented with anthocyanins from purple sweet potato and chitosan nanoparticles loaded with quercetin, observed excellent color variation properties dependent on pH, which were easily distinguishable to the naked eye. When subjected to fresh shrimp samples over a period of 5 days, the films exhibited a color variation from reddish pink to grayish purple. Additionally, the authors reported a significant improvement in antioxidant and antibacterial properties, enabling their application in food packaging.

Wagh et al. [24] evaluated multifunctional films based on cellulose nanofibers supplemented with extracts of red cabbage and its bio-residue, rich in anthocyanins, applying them to pork, fish, and shrimp samples to monitor their freshness in real-time. They observed visual color variations in the analyzed samples after 48 h of storage. According to the authors, these color variations occurred due to pH variations in the samples, ranging from pH 5.81 to 7.09 for fish samples, from pH 5.85 to 6.81 for pork samples, and from pH 6.27 to 7.20 for shrimp samples, with all samples considered deteriorated after 48 h.

## 5. Encapsulation as an Alternative Method for Incorporating Anthocyanins into Films

Due to their structure and molecular composition, anthocyanins have limited stability and are highly susceptible to degradation under adverse conditions, and pH variations, temperature, and the presence of oxygen are considered as barriers that hinder their application in the food industry [59]. Therefore, the implementation of innovative technological strategies is necessary to overcome the intense degradation of anthocyanins when they are used in films.

Among the most well-known technological processes, encapsulation involves loading a bioactive compound into capsules, forming a sphere that uniformly encloses the compound. In order to meet the application requirements of the compound to be encapsulated, the properties of the capsule can be altered according to the type of function the encapsulated active compound should provide to the final product, such as its composition, release mechanism, size, final shape of the capsule, and cost, with micro- and nano-encapsulation processes being the main targets of current research [60]. However, the development of an easy and effective strategy for anthocyanin encapsulation is currently necessary to improve its delivery through food-grade raw materials as carriers [61].

The most commonly used method for encapsulation is the spray-drying method, as it is a continuous process, cost-effective, and already has easily accessible and available equipment. However, it requires careful selection of the wall material to be used, as it must be compatible with the food product, possess mechanical resistance, an appropriate particle size, and thermal release or dissolution, among other factors, to produce the so-called microencapsulated particles [62]. When appropriate substances are used for encapsulation, the core products of the microcapsules can be protected from deterioration caused by adverse environmental conditions such as light, moisture, and oxygen [63].

In spray-drying microencapsulation, the most commonly used wall materials are carbohydrates such as maltodextrin, gum arabic, and emulsifying starches, which possess properties such as good solubility and low viscosity at high solids content, highly desirable characteristics for an encapsulating agent [64].

Other different techniques can be used for microencapsulation, such as spray chilling, spray cooling, extrusion, fluidized bed coating, centrifugal extrusion, freeze drying, coacervation, and interfacial polymerization, where the size of the particles typically varies between 1 and 5000 μm [65].

In addition, nanoencapsulation can be performed, where biopolymeric structures are enveloped with encapsulating materials with diameters ranging from 1 to 1000 nm [65]. Nanoencapsulation is a technique capable of enabling the use of compounds while overcoming losses during processing and storage, with its reduced molecular size (nanoscale) offering numerous advantages compared to isolated compounds or nanoparticles [59]. Figure 4 presents a simplified scheme of micro and nanocapsules.

In the literature, some studies can be found using micro and nanoencapsulation of anthocyanins process for the production of pH indicator films, aiming to improve their properties. Wu et al. [66] prepared chitin oxide nanocrystals to nanoencapsulate anthocyanins from purple cabbage, which enhanced the pH sensitivity, mechanical strength, and water barrier properties of glucomannan-based pH indicator films. This study demonstrated that nanocomposites formed by nanoencapsulation can significantly improve the barrier and mechanical properties of pH-sensitive indicator films.

Producing smart films based on chitosan and starch, combined with anthocyanins encapsulated by potato amylopectin nanoparticles, Zheng et al. [67] observed that, in addition to achieving an encapsulation efficiency of 89.7%, the nanoencapsulated anthocyanins exhibited a better color stability compared to anthocyanins without nanoencapsulation.

Jang and Koh [68] analyzed the encapsulation efficiency by freeze-drying anthocyanins extracted from the aronia fruit (*Aronia melanocarpa*) using combinations of maltodextrin with carboxymethyl cellulose (CMC), gum arabic, and xanthan gum as coating agents. According to the authors, the addition of polysaccharides increased the efficiency of anthocyanins from 65% to 87% when compared to using maltodextrin alone. Regarding stability and encapsulation efficiency together, the coating agents that showed the best results were CMC and xanthan gum, reinforcing their possible use in the food industry for formulations containing anthocyanins as coloring sources.

Although studies evaluating the effect of the micro and nanoencapsulation of anthocyanins on the final characteristics of pH indicator smart packaging can be found in the literature, their studies are still in their infancy, and the real effect of these processes when applied to food products should be better addressed. This is necessary to prove the actual efficiency of alternative methods in preserving the desired final characteristics of the packaging.

## 6. Conclusions and Future Perspectives

The importance of using smart packaging indicators has been widely explored, in addition to their application in the food industry. However, some points are still of great importance when formulating a film intended for monitoring the freshness of food. The mode of identification and action of the indicator film, whether by direct contact with the food or without contact, through pH variation or the detection of volatile compounds or gasses, is a crucial and paramount factor. Anthocyanin-based indicator films have shown great potential in monitoring widely consumed products, such as meat, fish, or dairy products, being able to act either in direct contact with food or by variations in the environment they are in, tracking food decomposition based on pH changes and consequently, its coloration. Its main advantage lies in being of natural origin, having a low cost and easy incorporation, and presenting color variation easily perceptible to the naked eye by the consumer. Additionally, the method of production and characterization of indicator films, including different polymers employed and their respective blends, reinforce the idea that in-depth studies are increasingly necessary, in which each of these components is varied.

However, some factors such as storage temperature and the presence or absence of light are still important for maintaining the stability of anthocyanins. In this context, the introduction of alternative techniques and methods aimed at increasing the stability of anthocyanins, such as the production of micro and nanoencapsulated particles, becomes the target of future studies with potential applications in the food industry. Further investigations are needed regarding the response of nanoencapsulated anthocyanins under real storage conditions and in different products such as meat, fish, and dairy derivatives, as well as the study of possible interactions and sensory alterations due to the contact of anthocyanins with food. Nevertheless, it can be said that anthocyanins present promising potential in the production of smart indicator packaging for food products with the aim of informing consumers quickly and effectively about the quality and freshness of the food contained therein.

## Figures and Tables

**Figure 1 polymers-16-02886-f001:**
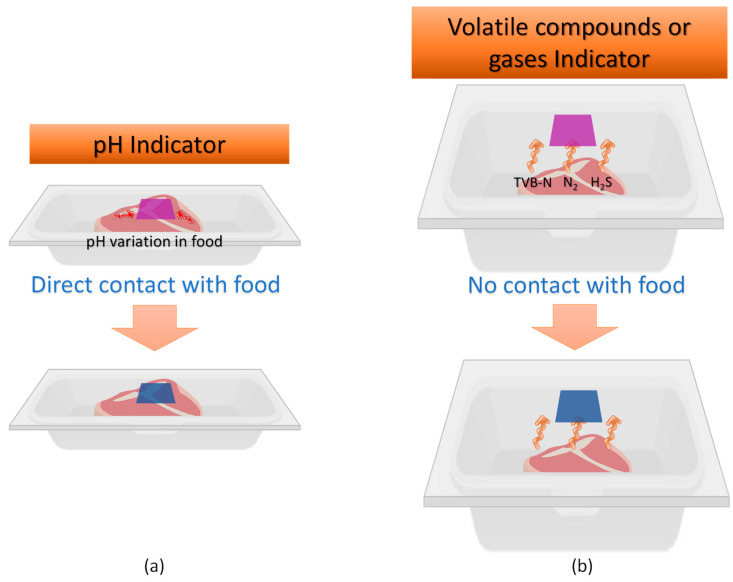
Schematic illustration of the color change mechanism of smart pH packaging film in case of (**a**) direct contact with the food, and (**b**) indirect contact with food.

**Figure 2 polymers-16-02886-f002:**
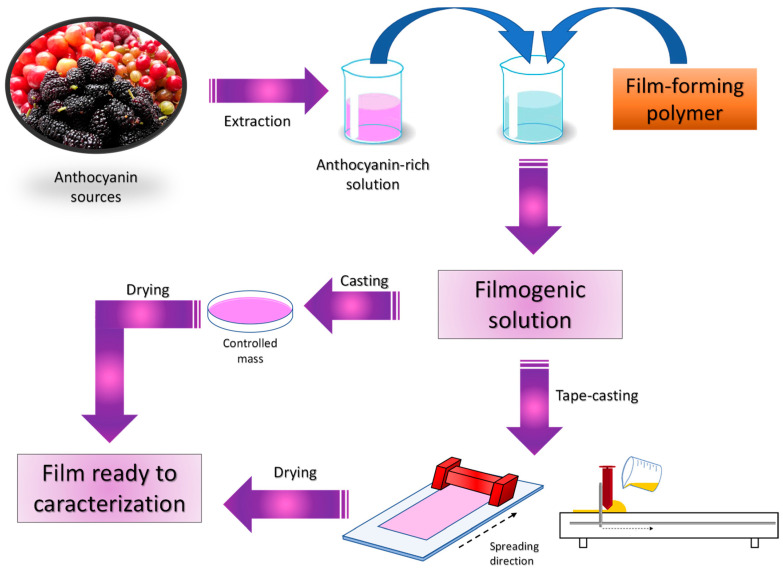
Scheme of the production of pH indicator films by casting and tape casting.

**Figure 3 polymers-16-02886-f003:**
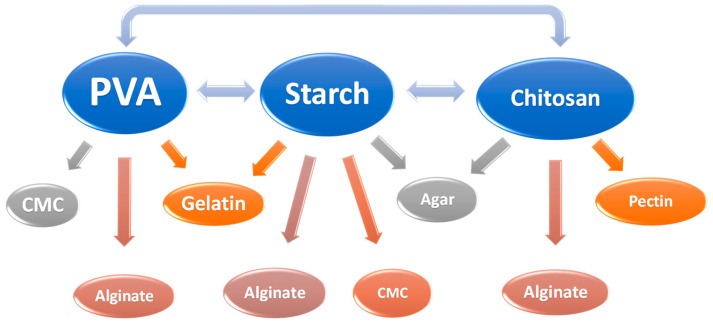
Combinations between polymers found in pH-sensitive smart films combined with anthocyanins.

**Figure 4 polymers-16-02886-f004:**
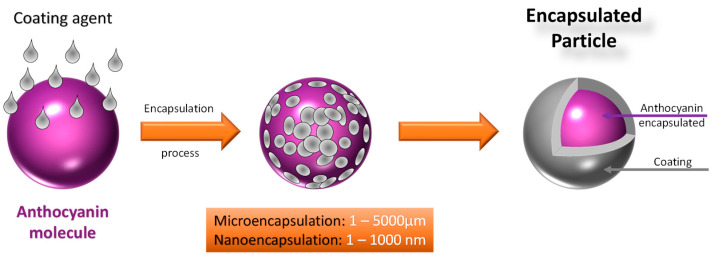
Scheme of microencapsulation and nanoencapsulation.

**Table 1 polymers-16-02886-t001:** Different sources of anthocyanins to produce pH indicator packaging with applications in foods and their characterizations.

Source of Anthocyanin	Film-Forming Polymer	Extract/Polymer Ratio	Food Application	Characterization	References
Red cabbage	PVA (1 g/100 g solution) and chitosan (1 g/100 g solution)	25 g/100 g polymer	Pasteurized milk	Swelling Index, Fourier-transform infrared analysis (FT-IR), differential scanning calorimetry (DSC), thermogravimetric analysis (TGA), mechanical properties, color parameters, sensitivity to lactic acid.	[20]
	Chitosan (1 g/100 mL solution) and corn starch (5 g/100 mL solution)	5 mL/100 mL solution	Fish filets	FT-IR, DSC, water vapor permeability (WVP), Light Microscopy (LM) and image texture analysis (GLCM and SDBC), color parameters, pH stability in different storage temperatures (4–7 °C and 25 °C for 72 h)	[21]
	PVA (8 g/100 g solution) and CMC (2 g/100 g solution)	0, 20, 40, and 60 mg/100 g solution	Fresh pork	Color parameters, mechanical properties, water solubility, swelling Index, FT-IR, X-ray diffraction (XRD), SEM, ammonia sensitivity, pH, and TVB-N.	[22]
	Pectin (2.5 g/100 g solution) and sodium alginate (2.5 g/100 g solution)	50.74 mg/100 g solution	Shrimp	Colorimetric response at different pH (2–12), UPLC, FT-IR, TGA, AFM, thickness, color parameters, UV-vis transmission, opacity, moisture content, water solubility, mechanical properties, antioxidant activity (DPPH and ABTS), color responses to pH (2–12), color stability in different storage temperatures (4 and 25 °C for 2 days), pH variation with and without contact with shrimp (4 and 25 °C for 72 h).	[23]
	Cellulose nanofiber (2.75 g/100 g solution)	6 g/100 g polymer	Pork, fish, and shrimp	FT-IR, X-ray photoelectron spectroscopy (XPS), SEM, AFM, transmission electron microscopy (TEM), antioxidant activity (DPPH and ABTS), antibacterial analysis (*Listeria monocytogenes* and *E. coli*), thickness, mechanical properties, color parameters, WVP, contact angle, TGA, ammonia sensitivity (NH_3_).	[24]
Purple sweet potato	Agar and potato starch	-	Pork	UV-vis spectroscopy, FT-IR, color response, pH variation for 48 h.	[7]
	Chitosan (2 g/100 g solution)	0, 5, 10 and 15 g/100 g polymer	-	Color response, thickness, moisture content, water solubility, WVP, UV-vis light barrier, mechanical properties, TGA, SEM, FT-IR, XRD, antioxidant activity (DPPH), color response to pH (3–10).	[6]
	Starch (4.5 g/100 g solution) and gelatin (4.5 g/100 g solution)	0.1 g sweet potato powder/100 g solution	*Flammulina velutipes* mushroom	FT-IR, SEM, color parameters, thickness, mechanical properties, moisture content, WVP.	[10]
Black carrot	Chitosan (2 g/100 g solution) and PVA (1 g/100 g solution)	1 g/100 g polymer	-	UV-vis spectroscopy, SEM, XRD, FT-IR, TGA, mechanical properties, color response to pH (3–13), color parameters, WVP, antibacterial analysis (*S. aureus*, *E. coli* and *P. aeruginosa*)	[25]
	Chitosan (1.4 g/100 g solution)	10.5 mg/100 mL solution	Pasteurized milk	Color response to pH (2–11), swelling Index, water solubility, FT-IR, SEM, color stability (20 °C and 30 days).	[26]
	PVA (3 g/100 g solution) and CMC (1.5 g/100 g solution)	45 mg/100 mL solution	Fish (Carp)	Color parameters, thickness, mechanical properties, moisture content, water solubility, WVP, light transmittance and opacity, TGA, SEM, FT-IR, XRD, antioxidant activity (DPPH), antibacterial analysis (*S. aureus* and *E. coli*), color response to pH (2–13), ammonia sensitivity (NH_3_), pH stability in different storage temperatures with and without light (4 and 25 °C for 20 days), total volatile basic nitrogen analysis (TVB-N)	[8]
Blueberry	Potato starch (6 g/100 g solution)	0.1 g/100 g polymer	Fresh shrimp	SEM, FT-IR, XRD, mechanical properties, water solubility, color parameters, ammonia sensitivity (NH_3_), color stability in different storage temperatures with and without light (4 and 25 °C for 14 days), color response to pH (2–12), microbial analysis by total viable count (TVC), TVB-N.	[27]
	Pectin (0.6 g/100 g solution), sodium alginate (0.5 g/100 g solution) and xanthan gum (0.2 g/100 g solution)	0.25, 0.5, 0.75 and 1.0 g/100 g polymer	Blueberry	Extract color response to pH (2–13), color response, thickness, mechanical properties, light transmittance and opacity, moisture content, water solubility, swelling Index, WVP, FT-IR, SEM, color response to pH (2–12), color stability in different storage temperatures (4 and 25 °C for 24 days), freshness in different temperatures (−1, 4, 10 and 15 °C)	[28]
Purple corn	Chitosan (2 g/100 g solution)	2 g/100 g polymer	-	FT-IR, color parameters, UV-vis transmission, thickness, moisture content, water solubility, WVP, mechanical properties, antioxidant activity (DPPH), antibacterial analysis (*E. coli*, *Salmonella*, *S. aureus* and *L. monocytogenes*), color response to pH (3–10).	[29]
Mulberry	PVA (10 g/100 g solution) and chitosan (3 g/100 g solution)	10, 20, 30, and 40 g/100 g polymer	Fish	TEM, SEM, color parameters, mechanical properties, UV-vis spectroscopy, color response to pH (1–11), changes in color parameters over 4 days.	[19]
	CMC (2.5 g/100 g solution)	10, 30 and 50 g/100 g polymer	Cherry tomato	Thickness, moisture content, water solubility, color parameters, opacity, UV-vis transmission, total phenolics, antioxidant activity (DPPH and FRAP), color response to pH (2–13), color stability (6 °C for 15 days).	[30]
Black rice	Chitosan (2.7 g/100 g solution) and pectin (1.3 g/100 g solution)	0.1, 0.2, 0.3 and 0.4 g/100 g polymer	Pork and beef meat	Antioxidant activity (DPPH), mechanical properties, FT-IR, SEM, color response to pH (1–13), ammonia sensitivity (NH_3_).	[31]
Roselle	Chitosan (1 g/100 g solution) and PVA (0.5 g/100 g solution)	0.02, 0.20, 0.33 and 0.46 g/100 g polymer	Fish filet	UV-vis spectroscopy, total phenolics and flavonoids, thickness, XRD, SEM, energy dispersive X-ray (EDX), moisture content, water solubility, WVP, oxygen permeability, contact angle, mechanical properties, TGA, release of anthocyanin, color response to pH (1–14), antioxidant activity (DPPH), cell viability, antibacterial analysis (*E. coli* and *S. aureus*), pH variation in fish (52 h), TVB-N (52 h).	[32]
	CMC (2 g/100 g solution), starch (2 g/100 g solution) and gellan gum (1 g/100 g solution)	-	Chicken breast	Anthocyanin stability in different temperatures (45, 60, 75 and 90 °C for 2 h), oxidation stability, light stability (constant light for 7 days) and pH stability (1–11), SEM, FT-IR, XRD, thickness, mechanical properties, moisture content, water solubility, WVP, oxygen permeability, opacity, antibacterial analysis (*S. aureus* and *E. coli*), sensibility response to pH (2–13), ammonia sensitivity (NH_3_), TVB-N.	[33]
Black eggplant	Chitosan (2 g/100 g solution)	1, 2, and 3 g de extract/100 g polymer	Pasteurized milk	Color parameters, thickness, moisture content, WVP, UV–vis light barrier, mechanical properties, SEM, FT-IR, XRD, antioxidant activity (DPPH), sensibility response to pH (2–13).	[34]

**Table 3 polymers-16-02886-t003:** Colorimetric response of pH indicator films when applied to different foods.

Application	Source of Anthocyanin	Film-Forming Polymer	Storage Time	Storage Temperature	Color Variation (Response)	References
Pork meat	Purple sweet potato	Agar and purple sweet potato	48 h	25 °C	Red to green (pH 5.78 to 7.46)	[7]
	Roselle	PVA, chitosan and starch	72 h	25 °C	Red to yellow (TVB-N 7.52 to 41 mg/100 g)	[43]
	Red cabbage	PVA and CMC	24 h	25 °C	Purple to blue purple (pH 5.71 to 6.51 and TVB-N 6.36 to 18.78 mg/100 g)	[22]
	Blueberry	PVA	5 days	4 °C 25 °C	Purplish red to dark blue (-)	[15]
	Black corn	Black corn starch and κ-carrageenan	23 h	25 °C	Pink to grayish green (TVB-N 2.70 to 14.03 mg/100 g)	[37]
	Black soybean	Sodium alginate	5 days	25 °C	Light brown to brown (pH 6.03 to 7.6 and TVB-N 9.20 to 33.98 mg/100 g)	[18]
	Eggplant	Chitosan	48 h	25 °C	Blue to green (pH 7.68 to 8.18 and TVB-N 15.16 to 24.32 mg/100 g)	[44]
Milk	Black cabbage	PVA and chitosan	4 days	25 °C	Dark gray to dark pink (pH 6.8 to 4.6)	[20]
	Black carrot	Chitosan	48 h	20 °C	Blue to violet rose (pH 6.6 to 5.7)	[26]
	Eggplant	Chitosan	16 h	40 °C	Blue to dark blue (pH 6.68 to 4.47)	[34]
Fish	Red cabbage	PVA and chitosan	72 h 7 days	25 °C 4–7 °C	Transparent to yellow	[21]
	Blueberry	PVA and chitosan	4 days	25 °C	Red to green (-)	[19]
	Black carrot	Bacterial cellulose nanofibers	15 days	4 °C	Deep carmine to jelly bean blue (pH 6.36 to 7.22)	[36]
	*Jacaranda cuspidifolia* flower	Chitosan and PVA	52 h	25 °C	Transparent to yellow (pH 6.5 to 12.8 and TVB-N 6.01 to 36.21 mg/100 g)	[32]
	Red cabbage	Gelatin	6 days	4 °C	Pink to atrovirens (TVB-N 13.30 to 37.76 mg/100 g)	[13]
Shrimp	Red cabbage	Pectin and sodium alginate	72 h	4 °C 25 °C	lilac to dark green (-) lilac to greenish yellow (-)	[23]
	Blueberry	Potato starch	36 h	4 °C	pink to light gray (pH 6.91 to 7.47 and TVB-N 9.57 to 28.47 mg/100 g)	[27]
Mushroom	Sweet potato	Starch and gelatin	60 h	20 °C	green to yellowish green (-)	[10]
Mitten crab	Mulberry	Chitosan, PVA, sodium alginate and pullulan	12 days	4 °C	Pink to dark green (pH 6.34 to 7.22 and TVB-N 9.2 to 45.24 mg/100 g)	[42]
Cherry tomato	Blackberry	CMC	15 days	6 °C	Red to reddish rose (pH 4.2 to 4.7)	[30]
Chicken	Purple sweet potato	Sodium alginate	60 h	4 °C 25 °C	Pink to blue (TVB-N 5.35 to 19.93 mg/100 g) Pink to blue (5.35 to 16.19 mg/100 g)	[16]
	Purple cabbage	Agar and methylcellulose	42 h	25 °C	Pink to green (TVB-N 8.8 to 44.0 mg/100 g)	[11]

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
