# Peer review of "Intelligent Packaging Systems with Anthocyanin: Influence of Different Polymers and Storage Conditions"

_polymers, 2024, doi:10.3390/polym16202886_

Round 1
Reviewer 1 Report
Comments and Suggestions for Authors
Anthocyanins have been widely used to prepare different intelligent films or packages. This review provides a sight of anthocyanins in indicator films with real applications in food products. The topic is meaningful. However, the poor writing and expression should be improved. There are two much data and results from references induced. Too much details about the results of other reports were used. Authors usually introduce the results from one report with whole paragraph. It is very important to summarize, think about the research progress. There are also some suggestions as follow:
1. Ln10-11, ‘the color change of colorimetric indicator films, not only due to changes in the surface of the food’, the change of whole food leads to the color change, such as volatile compound release.
2. Although ‘the influence of different polymers’ was mentioned in the manuscript title and abstract. The comparison, analysis and discussion of different polymers can not be found in the manuscript.
3. Ln 42’ those that changes their color based on pH variation have been widely studied in the literature in recent years’ please reorganize this sentence
4. Ln 54 please replace ‘packaging’ with ‘packages’
5. Ln 85, please list more reference to indicate many studies have demonstrated…
6. Ln 130, reorganize the sentence” Using milk samples, fresh cow milk is an amphoteric solution that is mainly related to the presence of citric acid, phosphates and casein..”
7. Ln 201 delete ‘can’
8. Ln 310’ When exposed to NH3, Zhang et al. observed significant color variations in polyvinyl alcohol (PVA) based indicator films added with blueberry extract.’ Please check the grammar. Who is exposed to NH3, do authors mean Zhang et. Al exposed to NH3 and observed?
9. The detail results and analysis from references are necessary to be introduced very much. Authors list to much data and results from references. Such as data and results from Bao et al. [27] has been introduced over whole three paragraph Ln 177-184, Ln 411-421 and Ln428-436. Data from Wu and Li [8] has been introduced over whole two paragraph Ln 158-167, Ln 396-404 and Ln 437-446.
10. Ln 609 ‘Alternative methods for incorporating anthocyanins into films’ please reconsidered the title. Many alternative methods should be introduced in this part. Actually, only encapsulation was mentioned here.
Comments on the Quality of English LanguageAnthocyanins have been widely used to prepare different intelligent films or packages. This review provides a sight of anthocyanins in indicator films with real applications in food products. The topic is meaningful. However, the poor writing and expression should be improved. There are two much data and results from references induced. Too much details about the results of other reports were used. Authors usually introduce the results from one report with whole paragraph. It is very important to summarize, think about the research progress. There are also some suggestions as follow:
1. Ln10-11, ‘the color change of colorimetric indicator films, not only due to changes in the surface of the food’, the change of whole food leads to the color change, such as volatile compound release.
2. Although ‘the influence of different polymers’ was mentioned in the manuscript title and abstract. The comparison, analysis and discussion of different polymers can not be found in the manuscript.
3. Ln 42’ those that changes their color based on pH variation have been widely studied in the literature in recent years’ please reorganize this sentence
4. Ln 54 please replace ‘packaging’ with ‘packages’
5. Ln 85, please list more reference to indicate many studies have demonstrated…
6. Ln 130, reorganize the sentence” Using milk samples, fresh cow milk is an amphoteric solution that is mainly related to the presence of citric acid, phosphates and casein..”
7. Ln 201 delete ‘can’
8. Ln 310’ When exposed to NH3, Zhang et al. observed significant color variations in polyvinyl alcohol (PVA) based indicator films added with blueberry extract.’ Please check the grammar. Who is exposed to NH3, do authors mean Zhang et. Al exposed to NH3 and observed?
9. The detail results and analysis from references are necessary to be introduced very much. Authors list to much data and results from references. Such as data and results from Bao et al. [27] has been introduced over whole three paragraph Ln 177-184, Ln 411-421 and Ln428-436. Data from Wu and Li [8] has been introduced over whole two paragraph Ln 158-167, Ln 396-404 and Ln 437-446.
10. Ln 609 ‘Alternative methods for incorporating anthocyanins into films’ please reconsidered the title. Many alternative methods should be introduced in this part. Actually, only encapsulation was mentioned here.
Author Response
Comments 1 - Ln10-11, ‘the color change of colorimetric indicator films, not only due to changes in the surface of the food’, the change of whole food leads to the color change, such as volatile compound release.
Response 1: We agree. It was changed to emphasize this point. The sentence has been modified to (Ln 10-12):
"Among intelligent packaging, colorimetric indicator films, which change color in response to changes in the food, such as the release of volatile compounds, have been widely studied."
Comments 2 - Although ‘the influence of different polymers’ was mentioned in the manuscript title and abstract. The comparison, analysis and discussion of different polymers can not be found in the manuscript.
Response 2: Thank you for your comment. The comparison, analysis, and discussion of different polymers are addressed in Section 4, where we discuss the production and characterization of indicator films. We discussed the results related to the color variation of these films.
Comments 3 - Ln 42’ those that changes their color based on pH variation have been widely studied in the literature in recent years’ please reorganize this sentence
Response 3: Thank you for pointing this out. We have revised the sentence to improve clarity and readability. The revised sentence now reads (Ln 43-44):
"Colorimetric indicator films that change color in response to pH variations have been extensively studied in recent years [5-9]."
Comments 4 - Ln 54 please replace ‘packaging’ with ‘packages’
Response 4: Thank you for pointing this out. As suggested by another reviewer, we have revised the entire sentence to (Ln 55-57):
“Numerous studies have utilized anthocyanins to produce different intelligent packaging films or smart labels to inform consumers about the quality and freshness of food.”
Comments 5 - Ln 85, please list more reference to indicate many studies have demonstrated…
Response 5: Thank you for your comment. The manuscript includes extensive references in Table 1 demonstrating the use of anthocyanins in pH indicator packaging and intelligent packaging systems for monitoring food quality and freshness. We have revised the paragraph to clearly reflect this and ensure there is no ambiguity. The updated sentence now reads (Ln 89-91):
“Recently, many studies have demonstrated the effectiveness of using anthocyanins in pH indicator packaging to monitor food quality and freshness, with the deterioration of many foods correlated with pH variations, as can be seen in Table 1.”
Comments 6 - Ln 130, reorganize the sentence” Using milk samples, fresh cow milk is an amphoteric solution that is mainly related to the presence of citric acid, phosphates and casein..”
Response 6: Thank you for pointing this out. We have reorganized the sentence for improved clarity. The revised sentence now reads (Ln 129-131):
“Fresh cow milk, which is an amphoteric solution mainly due to the presence of citric acid, phosphates, and casein, has a pH value between 6.6 and 6.8 and is one of the most commonly used parameters for analyzing milk quality [26].”
Comments 7 - Ln 201 delete ‘can’
Response 7: Thank you for your comment. We have removed the word "can" from Ln 199 as suggested.
Comments 8 - Ln 310’ When exposed to NH3, Zhang et al. observed significant color variations in polyvinyl alcohol (PVA) based indicator films added with blueberry extract.’ Please check the grammar. Who is exposed to NH3, do authors mean Zhang et. Al exposed to NH3 and observed?
Response 8: Thank you for pointing this out. We have revised the sentence to clarify it. The revised sentence is (Ln 305-308):
“When PVA-based indicator films added with blueberry extract were exposed to NH3, Zhang et al. [15] observed significant color variations. After just a 12-minute exposure period, the films exhibited color changes ranging from purplish-gray to blue and then to bluish-green.”
Comments 9 - The detail results and analysis from references are necessary to be introduced very much. Authors list to much data and results from references. Such as data and results from Bao et al. [27] has been introduced over whole three paragraph Ln 177-184, Ln 411-421 and Ln428-436. Data from Wu and Li [8] has been introduced over whole two paragraph Ln 158-167, Ln 396-404 and Ln 437-446.
Response 9: Thank you for your comment. We have reviewed the manuscript and consolidated the data and results from Bao et al. [27] (Ln 400-401 and Ln 413-416) and Wu and Li [8] (Ln 158-159, Ln 391-393 and Ln 419-423) to avoid unnecessary repetition. The revised sections now present the results and analyses in a more concise and focused manner.
Comments 10 - Ln 609 ‘Alternative methods for incorporating anthocyanins into films’ please reconsidered the title. Many alternative methods should be introduced in this part. Actually, only encapsulation was mentioned here.
Response 10: Thank you for pointing this out. To accurately reflect the focus of the section, we have revised the title of section 5 to (Ln 587):
"5. Encapsulation as an alternative method for incorporating anthocyanins into films."
Reviewer 2 Report
Comments and Suggestions for Authors
1- Among colorimetric indicator films, those that changes their color 42 based on pH variation have been widely studied in the literature in recent years. correct the grammar (change). Also, provide recent references to support your claim at the end of the sentence.
2- "Numerous studies use them to produce different intelligent films.." Please correct the grammar and the sentence " Numerous studies have utilized anthocyanins to produce different intelligent packaging films or smart labels ...."
3- "Thus, this review focus to provide an updated overview of current researches employing anthocyanins from different sources in indicator films with real applications in food products"
Please correct the grammar and the spilling mistakes as follows: "Thus, this review focuses on providing an updated overview of current research employing anthocyanins from different sources in indicator films with real applications in food products"
4- "Figure 1. Theoretical scheme of pH indicator packaging and volatile compounds or gases indicator packaging."
Pls. improve the caption as follows:
"Figure 1: Schematic illustration for the color change mechanism of smart pH packaging film in case of (a) direct contact with the food, and (b) indirect contact with food". Please add a/ b labels to Figure 1 and refer to them in the text.
5- references 20 and 21 are not recent! This does not reflect the state-of-art of the research subject. Pls. add recent articles related to red cabbage, the year 2020 onwards......
6- Under section 3, Please address the leaching out of anthocyanin from films. What strategies have been used to prevent/reduce this effect? for example, composites, crosslinking, etc... Please provide proper and recent references related to colorimetric/optical sensing of food freshness.
7- Under section 5, please provide some examples of nanocomposites for making pH-responsive films for food freshness monitoring to reflect the importance of mechanical properties in the prepared films. Please provide proper and recent references related to colorimetric/optical sensing of food freshness.
8- Provide an introductory section about the main types of anthocyanins, their colors, etc.., preferably with a supporting figure. especially those that appeared in this review. To help you, you may refer to this excellent review: ACS Food Science & Technology 1.2 (2021): 124-138
9- The title of section 4.1 does not reflect the content!!!! please check and correct.
10 - It is recommended that some figures from the discussed text be provided as visual aids to the review and attract the readers.
11- The English must be checked and the manuscript must be proofread.
Comments on the Quality of English Language
Minor/moderate English. I have provided few examples in my report.
Author Response
Comments 1- Among colorimetric indicator films, those that changes their color 42 based on pH variation have been widely studied in the literature in recent years. correct the grammar (change). Also, provide recent references to support your claim at the end of the sentence.
Response 1: Thank you for your comment. We have corrected the sentence to (Ln 43-44):
"Colorimetric indicator films that change color in response to pH variations have been extensively studied in recent years [5-9]."
Additionally, we have added recent references to support this claim at the end of the sentence.
Comments 2- "Numerous studies use them to produce different intelligent films.." Please correct the grammar and the sentence " Numerous studies have utilized anthocyanins to produce different intelligent packaging films or smart labels ...."
Response 2: Thank you for pointing this out. We have corrected the grammar in the sentence as follows (Ln 55-57):
"Numerous studies have utilized anthocyanins to produce different intelligent packaging films or smart labels to inform consumers about the quality and freshness of food."
Comments 3- "Thus, this review focus to provide an updated overview of current researches employing anthocyanins from different sources in indicator films with real applications in food products"
Please correct the grammar and the spilling mistakes as follows: "Thus, this review focuses on providing an updated overview of current research employing anthocyanins from different sources in indicator films with real applications in food products"
Response 3: Thank you for your comment. We have made the requested change and corrected the grammar (Ln 57-59).
Comments 4- "Figure 1. Theoretical scheme of pH indicator packaging and volatile compounds or gases indicator packaging."
Pls. improve the caption as follows:
"Figure 1: Schematic illustration for the color change mechanism of smart pH packaging film in case of (a) direct contact with the food, and (b) indirect contact with food". Please add a/ b labels to Figure 1 and refer to them in the text.
Response 4: Thank you for your comment. We have revised the caption for Figure 1 (Ln 81-82) and added the labels (a) and (b) to the image to align with the updated caption.
Comments 5- references 20 and 21 are not recent! This does not reflect the state-of-art of the research subject. Pls. add recent articles related to red cabbage, the year 2020 onwards......
Response 5: Thank you for your feedback. We acknowledge that references 20 and 21 are not recent. However, these references were selected because they provide foundational insights and relevant data that are still pertinent to the current state of research. We have ensured that the references included are relevant to the topic and provide valuable context for understanding the development and applications of anthocyanin-based indicator films. We believe that the current references comprehensively cover the key aspects of the research subject.
Comments 6- Under section 3, Please address the leaching out of anthocyanin from films. What strategies have been used to prevent/reduce this effect? for example, composites, crosslinking, etc... Please provide proper and recent references related to colorimetric/optical sensing of food freshness.
Response 6: Thank you for your feedback. While the leaching of anthocyanins from films is an important topic, it is beyond the scope of this review. Addressing this issue in detail would significantly extend the review and shift the focus from the primary objective of summarizing recent advancements in colorimetric and optical sensing of food freshness. The current review focuses on the integration of anthocyanins in indicator films and their application in food freshness monitoring. We have included recent and relevant references related to these aspects. We appreciate your understanding and hope the review remains aligned with its intended focus.
Comments 7- Under section 5, please provide some examples of nanocomposites for making pH-responsive films for food freshness monitoring to reflect the importance of mechanical properties in the prepared films. Please provide proper and recent references related to colorimetric/optical sensing of food freshness.
Response 7: Thank you for your feedback. Section 5 title has been revised and changed to “Encapsulation as an Alternative Method for Incorporating Anthocyanins into Films.” (Ln 587)
In this review, we provide an overview of key advancements in encapsulation techniques, including micro and nanoencapsulation, which are crucial for enhancing the performance of pH-responsive films. Due to the broad scope of the topic and to maintain the review's focus, we have not extensively detailed all nanocomposite examples or their mechanical properties in this section. Our goal is to provide a concise introduction to recent research while ensuring the review remains coherent and focused, without extensively elaborating on each method, in order to maintain the overall flow of the review. We believe this approach aligns with the intent of the review and provides a clear introduction to the topic.
Comments 8- Provide an introductory section about the main types of anthocyanins, their colors, etc.., preferably with a supporting figure. especially those that appeared in this review. To help you, you may refer to this excellent review: ACS Food Science & Technology 1.2 (2021): 124-138
Response 8: We appreciate the suggestion to include an introductory section on the main types of anthocyanins and their colors. However, after careful consideration, we believe that adding such a section might extend the scope of the review beyond its intended focus. Our review aims to concentrate on recent advancements in the application of anthocyanins in pH indicator films, specifically highlighting innovations and practical implementations relevant to the field. Including a detailed introductory section on anthocyanins and their colors would require an extensive discussion that could shift the review’s focus away from its primary objective. This approach allows us to maintain the review’s coherence and focus while directing readers to additional resources for more detailed background information. We believe this aligns with the review’s goals and provides a balanced presentation of the topic.
Comments 9- The title of section 4.1 does not reflect the content!!!! please check and correct.
Response 9: Thank you for your comment. We have revised the title to better reflect the content of the section. The new title is "Types of polymers used in pH indicator films" (Ln 483) which more accurately describes the discussion on the various polymers utilized in the creation of pH indicator films.
Comments 10 - It is recommended that some figures from the discussed text be provided as visual aids to the review and attract the readers.
Response 10: We appreciate the suggestion to include additional figures to enhance the visual appeal of the review. We have carefully considered the current figures and their role in supporting the text. We believe that the existing figures already provide a comprehensive visual representation of the key concepts and data discussed. Adding additional figures may not be necessary as the current ones are intended to effectively convey the main points and maintain the coherence of the review. We are confident that the figures included are well-suited to illustrate the content and will support readers' understanding of the material.
Comments 11- The English must be checked and the manuscript must be proofread.
Response 11: Thank you for your comment. We have carefully proofread and corrected the manuscript to improve the English language and ensure it meets high standards of clarity and accuracy.
Round 2
Reviewer 1 Report
Comments and Suggestions for Authors
The manuscript has been improved.
Comments on the Quality of English LanguageThe manuscript has been improved.
Reviewer 2 Report
Comments and Suggestions for Authors
The authors have responded to the main points and suggestions. The manuscript can be accepted in its current form.